

# Reanalyses of Maskelyne's Tidal Data at St. Helena in 1761

Philip L. Woodworth[1] and John M. Vassie[1]

[1] National Oceanography Centre, Joseph Proudman Building, 6 Brownlow Street, Liverpool L3 5DA, United Kingdom

*Correspondence to*: P.L. Woodworth (plw@noc.ac.uk)

**Abstract.** The construction of an electronic data set of the tidal measurements made at St. Helena in 1761 by Nevil Maskelyne is described. These data were first analysed by Cartwright (1971, 1972) in papers which have importance within studies of changing tides. However, Cartwright's data files were never archived for the benefit of other researchers, demonstrating that 'old data' at risk can sometimes take the form of electronic rather than paper records. In the present paper, the newly digitised Maskelyne data have been reanalysed by several techniques in order to obtain an updated impression of whether the tide has changed at that location in over two and a half centuries. Our main conclusion, consistent with that of Cartwright, is that the major tidal constituent (M2) has changed little. However, the results of the various techniques demonstrate how difficult it is to obtain reliable conclusions for the smaller constituents.

**Keywords**

Data reanalysis; Tidal science; Changing ocean tides; St. Helena; Nevil Maskelyne



## 1 Introduction

Almost fifty years ago, David Cartwright investigated whether the ocean tide at St. Helena had changed since 1761, with his findings reported in two papers (Cartwright, 1971,1972). This was an interesting piece of work at the time, but has gained additional importance since then, given our present understanding that the ocean tide has been changing in recent decades in many parts of the world (Woodworth, 2010; Haigh et al., 2020).

The comparison of the modern and historical tides at St. Helena was made using a year of high and low water data at Ascension Island in 1958-9 in order to provide a reference tide for the use of the response method in analyses of the short tidal records available from St. Helena. Ascension lies 1300 km northwest of St. Helena but can be considered 'nearby' in the context of the ocean tide response to astronomical forcing. Cartwright himself had been one of the developers of the response method (Munk and Cartwright, 1966). The modern tide at St. Helena was determined by means of Cartwright's own tidal measurements there for 39 days in 1969, while the historical tide was calculated from measurements made by Nevil Maskelyne for over a month in 1761, both data sets analysed using the response method and the Ascension reference record.

A listing of Maskelyne's tidal measurements at St. Helena is given at the end of Maskelyne (1762) as shown in Figure 1. Although Cartwright must have spent a lot of time putting these measurements into electronic form, it is impossible for anyone to readily repeat his work now because he did not lodge his files in a data centre. Back in 1971 there was no culture of depositing data sets in centres such as the British Oceanographic Data Centre (BODC) or even of providing Supplementary Material for a paper.

'Data reanalysis' comes into this discussion because we wanted to see if we would obtain the same findings as Cartwright, should the Maskelyne data be made available electronically once again, especially given the present interest in changing tides (Haigh et al., 2020). As a result, a summary of our own conclusions on the tides at St. Helena, based on analyses of Maskelyne's data set, compared to those of Cartwright is given below.

## 1761.

| Day of ob-servation. | Apparent time | | The height in divisions and tenths. | | No. of observ-ations. | N. B. The height is set down according to the divisions on the post, each of which is 3 inches. |
|---|---|---|---|---|---|---|
| | h | l | | | | |
| Nov. 2⟑ 12. | 8 | 56 A. M. | 1 | 1/8 | L 17 | |
| | 9 | 32 A. M. | 1, | 8 | 34 | |
| | 9 | 52 A. M. | 2 | 1/4 | 18 | |
| | 10 | 37 A. M. | 4, | 0 | 16 | |
| | 0 | 58 P. M. | 11 | | 16 | |
| | 2 | 27 P. M. | 12, | 4 | H 55 | |
| | 3 | 29 P. M. | 12 | | 18 | |
| | 3 | 49 P. M. | 11, | 3 | 16 | |
| | 4 | 32 P. M. | 9, | 4 | 20 | |
| | 8 | 26 P. M. | 1 | | L | |
| | 9 | 54 P. M. | 2 | 1/2 | | |
| ⟑ 13. High surf. | 6 | 45 A. M. | 4, | 8 | 38 | |
| | 7 | 24 A. M. | 3, | 3 | 36 | |
| | 7 | 57 A. M. | 2, | 5 | 12 | |
| | 9 | 21 A. M. | 1, | 5 | L 12 | |
| | 9 | 32 A. M. | 2, | 0 | 20 | |
| | 1 | 26 P. M. | 10, | 9 | 28 | |
| | 3 | 21 P. M. | 13, | 3 | H 22 | |
| | 3 | 54 P. M. | 12, | 8 | 26 | |
| | 6 | 35 P. M. | 6, | 3 | 20 | |

**Figure 1.** The start of the table of measurements of sea level at St. Helena in 1761 to be found at the end of Maskelyne's original paper. Source: Maskelyne (1762) reproduced with permission of the Royal Society.

## 2 The Maskelyne Data Set

The first person to make tidal measurements at St. Helena was the Rev. Nevil Maskelyne, Astronomer Royal 1765-1811. Maskelyne's reason for visiting the island was to observe the time of the transit of Venus on 6 June 1761, an objective that was prevented by the cloudy weather. His exercise in tide recording, from 12 November to 22 December 1761, must have compensated somewhat for the failure of that main objective.

As explained by Cartwright et al. (2017), knowledge of the tide was still rudimentary in 1761. Newton had shown that the main characteristics of the ocean tide followed from his gravitational theory. However, there was a lack of observational data from which one could learn more about tidal dynamics, especially from remote island locations. St. Helena was certainly remote but had one major drawback, in that its exposure to swell waves made it difficult to observe the modest tidal rise and fall (the mean tidal range is approximately 1 metre).

Maskelyne managed to largely eliminate the effect of swell waves by reading from a graduated vertical staff many times over the course of a few minutes and recording the average reading: "I therefore generally made 40 or 50 observations, and sometimes more than 100, if the rise and fall of the water seemed very irregular". The resulting averages traced out a smooth tidal curve, and simultaneous readings by Maskelyne and his assistant Charles Mason agreed consistently to better than half an inch (12 mm). (Mason is better known as the leading surveyor of the Mason–Dixon Line and for the measurement of a degree of latitude in North America.) Their observations were made for all states of the tide between 12 November and 22 December 1761, except for a short interruption when the swell damaged the vertical staff. [1]

The resulting measurements can be found in a table at the end of Maskelyne (1762), and the present exercise involved the typing of those numbers into a single ASCII computer file. That file, called 'maskelyne_data', has a format which is essentially the same as Figure 1. It consists of spot measurements of heights at particular times, all of which come from Maskelyne's table apart from a couple of errors which Cartwright pointed out in a footnote at the bottom of Cartwright (1971, p617). In these cases, we have used the Cartwright numbers instead. The times corresponding to each tide level are given as hours and minutes and the levels themselves are in 'divisions and tenths' where one division is 3 inches. Also shown is the number of separate instantaneous estimates of height made rapidly by Maskelyne over several minutes, averaged and recorded to the nearest minute. There are some days which are almost complete, with measurements of tide level around the clock. However, as pointed out in Cartwright (1971), in the latter part of the data set the measurements are increasingly in daylight hours (Figure 2). Cartwright (1971) should be consulted for further explanation of how Maskelyne came to make his measurements and for additional details about them, while the header of 'maskelyne_data' contains more detailed information about the file itself. Section 6 mentions the locations from which this new data file might be obtained.

Before comparing our findings with those of Cartwright in the next sections, we can point to several remarks in his papers to do with the data that we have concerns about:

- On page 617 of Cartwright (1971) he says "Each sea level [measured by Maskelyne] was the mean of up to 100 or more observations at different states of the swell over several minutes, and recorded against the mean time to the nearest 1/4 min."

One can see from Maskelyne's table that there were certainly some sea level measurements obtained from over 100 separate observations. However, that was the case for only 10 of the measurements out of 478 in total. Normally, there were only a few 10s of observations. Therefore, it seems that this sentence of Cartwright (1971) over-states the quality of Maskelyne's data somewhat.

In addition, it is not clear where the "mean time to the nearest 1/4-min" statement came from. Maskelyne (1762, p589-590) says "I always looked at my watch before I began to note the height of the water, and looked at it again when I had finished the experiment; the medium of the two times I set down as the true time of the observation. The times set down are exact to the minute."

- The first footnote on Cartwright (1971, p617) says "I was unable to detect any sensible change in datum after the pole was re-set."

---

[1] The zero of Maskelyne's staff was not related to a land benchmark, so the historical data are not useful to studies of long-term sea level change.



This is a reference to the entry in Maskelyne's table showing that the tide pole was damaged on 3 December and
that Maskelyne put it back on 5 December saying "The post was set up again as near the former height as could be
judged." However, as discussed below, there is some evidence for a datum shift of about 2 or 3 cm either side of this
event, and this datum shift impacts significantly on determination of changes in the diurnal tides in particular.

• There are also some additional minor inconsistencies. For example, there were 40 days with measurements
as stated in Cartwright (1971, p617). They spanned 41 days (12 Nov – 22 Dec) with a day with no data on
4 December. However, in Cartwright (1972, p337) he implies that the span was 42 days. And in his table
on p338, he says he used only 39 days of data which 'exclude certain lacunae [gaps] in the data'. But the
only gap was the 4 December one. If he used only 39 days, it is not obvious which day of the 40 he dropped;
there were many days shown as having high surf which he might have been rejected otherwise.

• Finally, p617 contains a long paragraph concerning what the longitude was of the clock used by Maskelyne
and Mason. While this paragraph is interesting, it does not really seem relevant to the present comparison
of historical and modern tides. Even if there was an uncertainty of 0.1 ° in longitude as he suggests, that
propagates into uncertainties of only 0.1/0.2° in the calculation of the phase lags of diurnal/semidiurnal
tides which is well within any realistic uncertainty in a comparison.

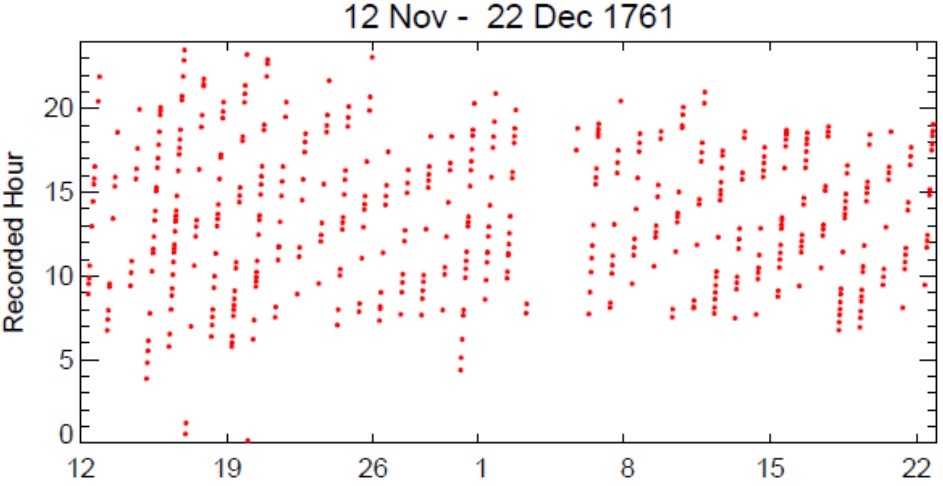


**Figure 2.** The recorded (local) times of Maskelyne's measurements. Some measurements were made around the clock in the earlier part of the data set. However, they can be seen to be restricted to daylight hours in the latter part.




# 3 Data Reanalyses

## 3.1 Time Corrections to Maskelyne's Measurements

Before a comparison can be made between the modern and historical tides, it is necessary to convert Maskelyne's times, which are local apparent (sun dial) times, to Greenwich Mean Time (GMT) in a similar way to Cartwright (1971). First, we adjusted for longitude using the same value (5.718 °W) used for the study of Manuel Johnson's data at St. Helena in 1826-1827 (Cartwright et al., 2017).[2] All such historical measurements of the tide at St. Helena have been made at the landing steps in Jamestown Bay, as they continue to be made to this day. Then, we corrected for the equation of time (EOT) using EOT values from the 1795 Nautical Almanac.

We used EOT values from the 1795 edition of the Almanac because we happened to have a copy available. The Nautical Almanac was first published in 1767 and so it did not exist in 1761, so we had to use EOT values for not too different a later year. Fortunately, 1761 and 1795 were both non-leap years and their EOT values should be very similar as any changes in the EOT over only three decades will be negligible; as a confirmation, we checked that the small differences in the EOT over two centuries between 1795 and 1991 in the tables we used were the same as those shown in Hughes et al. (1989). The resulting total time corrections for each day (i.e. longitude and EOT) were very similar to those listed in Table 5 of Cartwright (1971).

## 3.2 Initial Comparison of Modern and Historical Tides

In order to make our own comparison of the modern and historical tides, we initially made use of a set of 62 tidal constants derived from a record of sub-surface pressure (SSP) at St. Helena spanning one year (October 1995 - October 1996). This set is called STHL4. It contains constants for 5 long-period, 18 diurnal, 20 semidiurnal and 19 higher-frequency constituents. Although the record is of SSP and not real sea level, there is not much difference between the constants that would be obtained from the two as the product of density and acceleration due to gravity ($\rho g$) happens to be almost exactly 1.0 at St. Helena. This was confirmed by comparison to another set of constants for real sea level during 1993-2006 computed by Richard Ray for the Manuel Johnson study (Cartwright et al., 2017).

However, there was the expected difference for S2 (the main solar semidiurnal tide) which has an amplitude of 10.25 cm in STHL4 and 11.39 cm in Ray's set. That can be explained by the S2 air tide at St. Helena having an amplitude of 1.1 mbar and a phase lag which is almost opposite that of sea level (Ray, 1998). As a result, S2 in SSP has an amplitude about 1 cm smaller than in sea level.

Therefore, for present purposes we defined a new set of constants based on STHL4 but with those for S2 taken from Ray's set. For consistency, we also replaced those for S1 by Ray's although this has an amplitude of only 1.4 mm, consistent with Figure 3 of Ray and Egbert (2004). This set of 62 harmonic coefficients is called STHL4.X. The amplitudes (H) and Greenwich phase lags (G) of the five main constituents (TC) and their origins (see Pugh and Woodworth, 2014) are listed in Table 1.

One obvious thing to point out is how small the diurnal tides are at St. Helena and, therefore, how difficult it would be to decide reliably on any changes in them from one epoch to another, even if one had a longer historical data set than Maskelyne's measurements of over a month and with irregular timing. Our findings, and those of Cartwright (1971) to be discussed below, should be considered with this reservation in mind.

The STHL4.X harmonics were used to make 1-minute predictions of the tide for 1761 from which we picked out values at exactly the same times as Maskelyne's measurements.[3] These are called 'Predicted' values although 'Hindcasted' would probably be a better description (Cartwright's expression for them was a 'tidal synthesis'). There are then 478 of them corresponding to the same number of Maskelyne values.

Figure 3(a) shows the time series of 1-minute predicted heights from 12 November - 22 December together with the Maskelyne sea levels shown by red dots. The two sets of values have been adjusted to have zero mean. It can be

---

[2] It was not possible to include in the present study information on the tide at St. Helena based on the Manuel Johnson measurements in 1826-1827 as they were not of sufficient quality for examining small changes in the tide through the years; see Cartwright et al. (2017) for details.

[3] These predictions assume the same nodal variations for the lunar tides as in the equilibrium tide, which is a reasonable assumption for an ocean island location. The end of 1761 is anyway not at a time of nodal maximum or minimum, so any uncertainty arising from this assumption will be small.



seen that many of Maskelyne's measurements took place around high or low tide as was his intention (Maskelyne, 1762). However, there were also many measurements around mid-tide. Figure 3(b) focusses on a subset of data for 15-20 November, demonstrating the general good correspondence of predicted heights and Maskelyne's measurements.

Figure 3(c) shows sea level differences (Maskelyne - Predicted) with the overall mean difference removed. One can see that there is an apparent datum shift at the time that Maskelyne's tide pole was damaged on 3 December and replaced on 5 December. Determining the size of a datum shift is difficult when the shift is comparable to the variability in the record due to fluctuations in the ocean water properties (especially temperature) and to meteorological effects, and it is sometimes difficult even deciding if there is a shift at all. However, simple inspection suggests a shift of about 2.8 cm at that time, estimated from the difference between the average sea level differences either side of the gap. Figure 4 (a) shows that after adjustment for the shift the Predicted and Maskelyne sea levels values correspond satisfactorily (as in fact do the unadjusted vales given that 2.8 cm is a small amount compared to the tidal range). Figure 4(b) shows that the sea level differences have no major dependence on tidal level.[4]

An important issue at this point is that Cartwright (1971) did not believe that there was any evidence for a datum shift. Therefore, his findings were based on an analysis of the complete Maskelyne data set without consideration of either a datum shift or the possible importance of long-period tides (which in this case amounts to much the same thing). We made a considerable number of tests using predictions based on STHL4.X of whether findings on the tidal composition of Maskelyne's data could be affected by his irregular temporal sampling and/or by a datum shift and/or by long-period tides. There were too many tests to be described in detail in this short note but our general conclusion was that a datum shift of 2-3 cm or the presence of long-period tides (or not) would not impact significantly on the determination of the main semidiurnal tides (M2 and S2) but would be important for the diurnals, with uncertainties of about 10% in their amplitudes. These initial tests informed our choice of methods employed in the next sections.

**Table 1:** Amplitudes (H, cm) and Greenwich Phase Lags (G, deg) of the main tidal constituents (TC) in the STHL4.X set and their origins.

| TC | H (cm) | G (deg) | Origin |
|----|--------|---------|--------|
| M2 | 32.49 | 80.04 | Principal lunar semidiurnal |
| S2 | 11.39 | 101.96 | Principal solar semidiurnal |
| K1 | 3.46 | 349.22 | Principal lunar/solar diurnal |
| O1 | 2.08 | 190.65 | Principal lunar |
| N2 | 6.69 | 70.88 | Larger elliptical lunar semidiurnal |

---

[4] The apparent larger scatter at high and low waters than at mid-tide in Figure 4(a), which is counter intuitive as measurements are normally more accurate at the turning points than at mid-tide when the water level is changing rapidly, is an artefact of there being more measurements at the high and low water levels. The standard deviation of Maskelyne minus Predicted levels in Figure 4(b) is 3.2, 3.0 and 3.0 cm for bands of predicted level -60 to -20, -20 to 20 and 20 to 60 cm respectively.


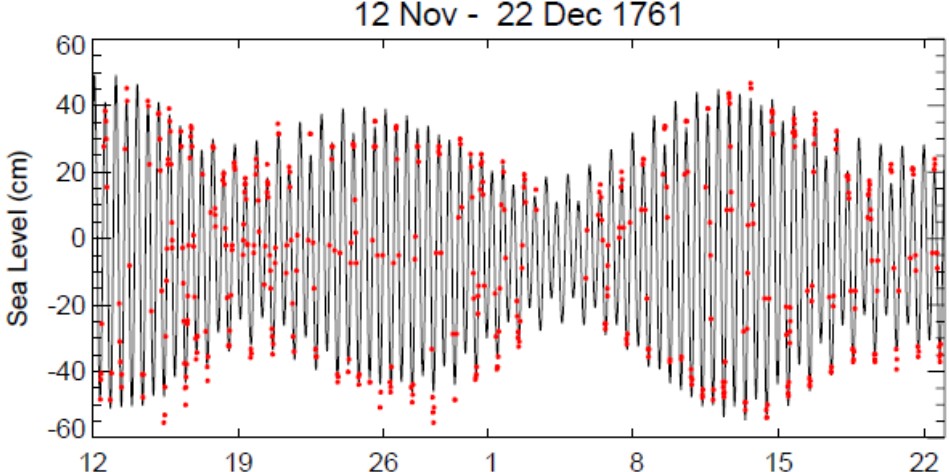

**Figure 3(a).** 1-minute predicted heights for 12 November - 22 December 1761 together with the Maskelyne sea
levels shown by red dots. The two sets of values have been adjusted to have zero mean.

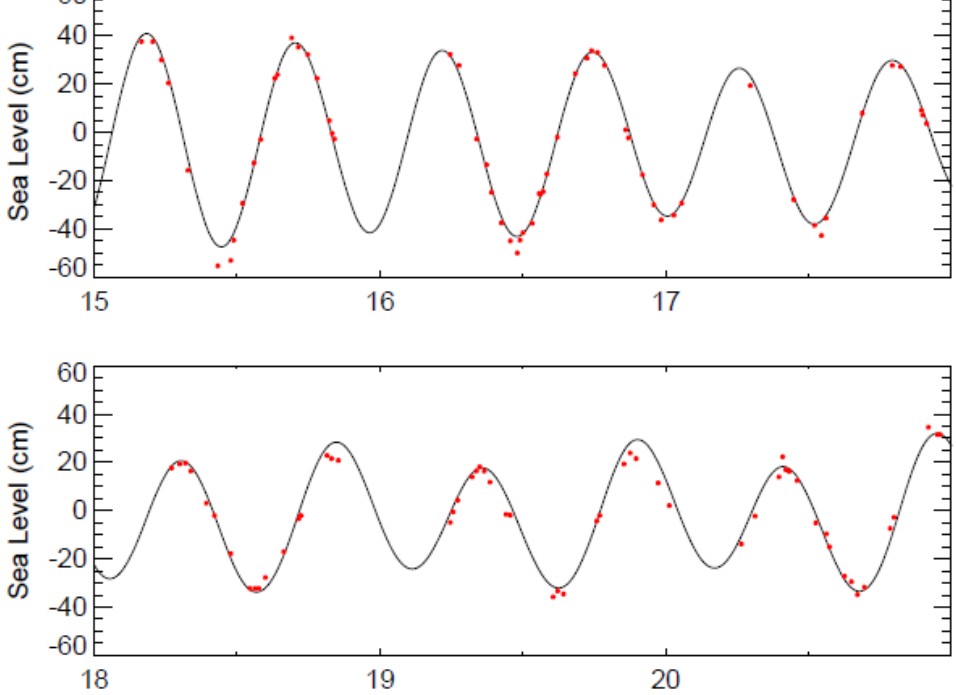

**Figure 3(b).** A subset of (a) focussing on 15-20 November 1761 which corresponds to Figure 4 of Cartwright (1971),
although the predicted tide will be slightly different in the two cases, and a couple of outlying Maskelyne's



measurements shown here appear not to have been used by Cartwright. Before and after the six days shown the
observations of Maskelyne are mostly confined to daylight hours.

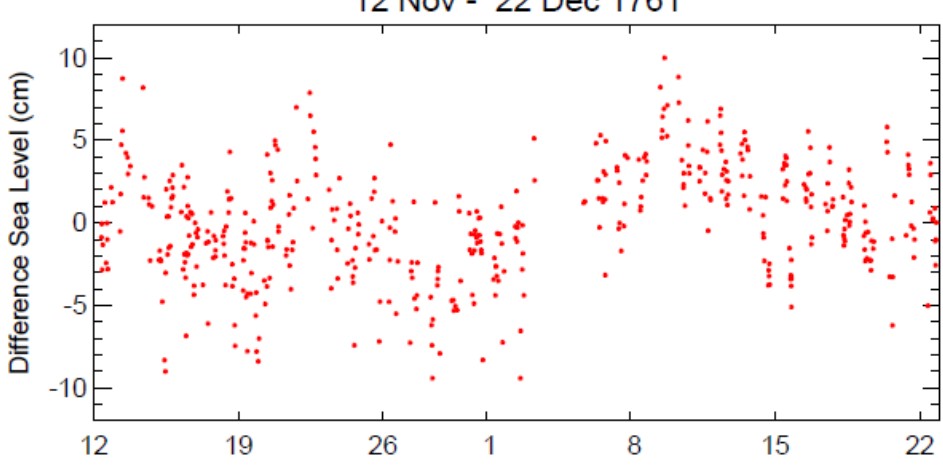

**Figure 3(c).** Sea level differences (Maskelyne - Predicted) with the overall mean difference removed.


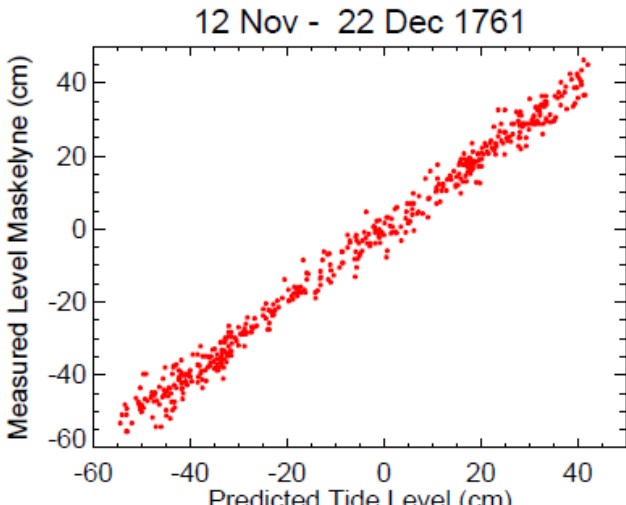

**Figure 4 (a).** Predicted and Maskelyne sea levels values after datum shift adjustment.

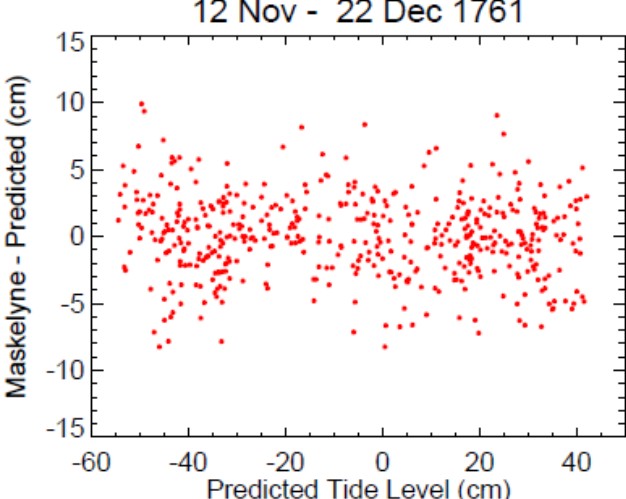

**Figure 4 (b).** Sea level differences (Maskelyne-Predicted) after datum shift correction showing no major dependence
on tidal level.



### 3.3 New Harmonic Tidal Analysis (First Method)

The next question is how to analyse more rigorously the modern and historical (i.e. Predicted and Maskelyne) measurements in order to see how the tide might have differed between the two epochs. To do that, a form of harmonic analysis was devised involving harmonic fitting to the two sets of 478 heights. The obvious drawback to such a fit is that the data are not regularly spaced in time, so an initial concern was that fits could be biased towards the earlier days with many measurements. However, the residuals of the fits described below look much like those of Figure 3(c), with similar earlier and later residual variances. Therefore, this seems not to be the case.

Several variants of harmonic fits were tried and what seems to be the best is described here. This considered just five main constituents (M2, S2, K1, O1 and N2) as it was believed that the available data justified using no more than that. N2 was included in its own right, rather than as a related constituent, because it is so large (amplitude of almost 7 cm, and larger than either K1 and O1) and in theory it should be determined adequately with a month of data.

Several other harmonics were included as related constituents. These included K2 and T2 related to S2, P1 related to K1, Q1 related to O1, and NU2, 2N2 and MU2 related to N2. Each of the related constituents were required to have amplitudes relative to the main constituents in proportion to those in STHL4.X, and differences in phase lags to those of the main constituents also as in STHL4.X, but with related amplitudes and phase lags adjusted for the nodal cycle as appropriate for the end of 1761.

Various tests were made to ensure: (i) that the method and program code perform as required, and (ii), as mentioned above, that the irregular sampling of the 478 Maskelyne times does not in itself unduly bias the determined amplitudes and phases for the main constituents, especially the semidiurnals. The fit thereby contains 10 parameters for the five main constituents and an optional 11[th] parameter to represent the possible datum shift on 4 December.

Changes were made before making these fits to both the Predicted and Maskelyne data considered previously in Section 3.2, in order to remove the complications of long-period tides from both of them. This involved a simple change for the Predicted data, in that we used only 57 of the 62 harmonics in STHL4.X (i.e. without the 5 long-period components Sa, Ssa, Mm, MSf and Mf).

For Maskelyne's data, the contributions of Mm, MSf and Mf to the measurements were removed using values from the Finite Element Solution (FES) 2014 model (Lyard et al., 2021). Of these, Mf is the most important and MSf the least important. The model has H (cm) and G (deg) values of 1.49 and 358.64 for Mf, 0.69 and 355.46 for Mm, and 0.020 and 209.15 for MSf respectively. Corrections for the seasonal constituents Sa and Ssa were not made as it was decided that they were unlikely to be important. Eight years of monthly mean sea level (MSL) values for St. Helena in the Permanent Service for Mean Sea Level (PSMSL) data set (Holgate et al., 2013) indicated that November levels were on average only 11 mm above December levels but could vary between +36 and -15 mm. Therefore, given that we also had no way to correct for daily and weekly changes in MSL, no monthly MSL adjustments were made.

The results when including the 11[th] parameter in the two fits to Predicted and Maskelyne data were as in Table 2(a). AR is the ratio of historical to modern amplitudes and PHLD is the historical phase lag minus modern phase lag. The fit to Predicted values suggested a possible datum shift of -0.2 cm, consistent with zero, as expected as of course the predicted time series has no shift in it. On the other hand, the fit to the historical Maskelyne values resulted in 1.4 cm for the possible datum shift. This is a smaller shift than obtained in the initial look at the data in Section 3.2; accounting for the long-period tides in the Maskelyne data using the FES2014 model values reduced its estimated amount.

The fits indicate that there has been no change to the main semidiurnal tides (M2 and S2) although N2 is suggested to be 15% smaller in the historical tide. On the other hand, they suggest that K1 had only 84% of its modern value in historical times, and there was a 12% larger historical O1. These departures of the amplitudes of the historical diurnals from their modern values of approximately 10% are consistent with the anticipated biases arising from the various tests with STHL4.X mentioned above.

However, if one believed there to be no datum shift, as in Cartwright (1971), then fits using 10 parameters only result in the values shown in Table 2(b). In this case, the M2 and S2 semidiurnal tides appear to be much the same in the two epochs, with S2 just a few percent smaller in the historical data. However, it results in historical K1 being only 72% of its modern value which is less plausible, while historical O1 is suggested to have been 8% larger. These





two sets of results demonstrate how sensitive the findings for the diurnals are to whether or not there was a small
datum shift, even if only centimetric.
**Table 2:** (a) Values of ratio of historical to modern amplitudes (AR) and historical phase lag minus modern phase
lag (PHLD, deg) when including an 11[th] parameter in the fits to Predicted and Maskelyne data by the first harmonic
method (Section 3.3). (b) Values of AR and PHLD when not including an 11[th] parameter in the first harmonic
method. (c) Values obtained using the second harmonic method (Section 3.4).
(a)

| | M2 | | S2 | | K1 | | O1 | | N2 |
|---|---|---|---|---|---|---|---|---|---|
| AR | PHLD | AR | PHLD | AR | PHLD | AR | PHLD | AR | PHLD |
| 0.994 | −0.50 | 0.992 | 1.263 | 0.841 | 1.758 | 1.124 | 9.613 | 0.846 | −0.463 |

337 (b)

| | M2 | | S2 | | K1 | | O1 | | N2 |
|---|---|---|---|---|---|---|---|---|---|
| AR | PHLD | AR | PHLD | AR | PHLD | AR | PHLD | AR | PHLD |
| 0.992 | −0.690 | 0.961 | 1.817 | 0.719 | −6.029 | 1.080 | 12.021 | 0.841 | −0.812 |

342 (c)

| | M2 | | S2 | | K1 | | O1 | | N2 |
|---|---|---|---|---|---|---|---|---|---|
| AR | PHLD | AR | PHLD | AR | PHLD | AR | PHLD | AR | PHLD |
| 0.997 | −0.48 | 1.040 | 2.35 | 0.938 | 8.23 | 0.952 | −5.38 | 0.935 | −1.4 |
| | | [0.951] | | | | | | | |

**3.4 New Harmonic Tidal Analysis (Second Method)**
As explained above, a problem with analysing the Maskelyne data set is the irregular temporal sampling of his
measurements. Therefore, in order to provide a more conventional time series for use in a second form of harmonic
analysis, use was made of the predicted tide in order to interpolate at 1-minute intervals between Maskelyne's
measurements, applying linear adjustments to the predictions between consecutive Maskelyne measurements so as
to correspond faithfully to the original data when available. There is obviously some danger in this approach,
particularly for calculation of the diurnal tides, given that many of the later Maskelyne measurements were during
daytime only. Therefore, there is a possibility of some information content from the predicted tide passing into the
interpolated Maskelyne tide. However, in spite of these reservations it was decided that this approach offered an
interesting alternative method. The predicted tide in this case was taken from a set of 62 harmonics called STHL2
calculated from an SSP record at St. Helena spanning November 1993 to February 1995. Its five main constituents
have amplitudes and phase lags as shown in Table 3. These STHL2 values are almost identical to those of STHL4,
but the amplitude of S2 is 0.914 of that in STHL4.X for the air tide reasons explained in Section 3.2.
**Table 3:** Amplitudes (H, cm) and Greenwich Phase Lags (G, deg) of the main tidal constituents (TC) in the STHL2
set.

| TC | H (cm) | G (deg) |
|---|---|---|
| M2 | 32.37 | 79.92 |
| S2 | 10.41 | 99.57 |
| K1 | 3.49 | 349.71 |
| O1 | 2.03 | 187.89 |
| N2 | 6.67 | 70.87 |

Using STHL2 results in a time series of 59040 predicted 1-minute values between 12 November and 22 December
1761, and a separate time series of Maskelyne values interpolated using STHL2. Each of these was analysed using



a conventional monthly tidal analysis containing 27 independent constituents including two long-period constituents
(Mm and MSf), and 8 related constituents with relationships to corresponding independent constituents taken from
those in STHL2. Findings were as shown in Table 2(c), again suggesting similar semidiurnal tides (M2 and S2) in
modern and historical times, although the AR value for S2 slightly larger than 1.0 converts to 0.951 once correction
for the different S2 amplitudes in SSP and real sea level are allowed for (shown by the square brackets). There were
smaller historical diurnal tides than modern ones and smaller N2.
This tidal analysis results in interesting findings for the two long-period tides. These are calculated to have
amplitudes (cm) for Mm of 3.143 (historical) and 0.577 (predicted), and for MSf of 2.597 (historical) and 2.016
(predicted). However, as noted above, the amplitude of the real Mm at St. Helena is only 0.69 cm, if one assumes
the FES2014 model to be correct. It turns out that the much larger amplitude of Mm obtained in the analysis of the
historical data than that of the predicted data, together with its phase lag, is consistent with simply parameterising
the possible datum shift discussed in Section 3.3 in a different way.
In summary, The results of this second harmonic analysis for the main semidiurnal tides are essentially identical to
those of the first harmonic method in Section 3.3. In addition, both suggest smaller historical diurnal tides than
modern ones, although this second harmonic method suggests more stable K1 and O1 than does the first method.

**3.5 New Response Analysis**

In a further type of tidal analysis, we attempted to reproduce the work of Cartwright (1971) by modifying the
response analysis software which Cartwright used for the analysis of short records.
Unfortunately, the version available to us did not work with randomly spaced data and had to be modified to do so
and rewritten in the Delphi language. It filters a reference time series with 6 bandpass filters: diurnal, semidiurnal
and ter-diurnal, each having a real and conjugate part. In addition, the original version of the software lags the
reference data by 2 days relative to the observations and applies the same filters. As a result, there are 12 band-
passed series in total. The software then calculates a co-variance matrix between a reference series and the data under
investigation, and response weights are calculated relative to these 12 series. For present purposes, we extended the
method to employ reference series that also lead the observations by 2 days, thereby making the analysis
symmetrical, resulting in 18 band-passed reference series in all. The data themselves were not filtered.
We applied the program to the Maskelyne data using St. Helena predictions as a reference, derived from STHL2 as
described above. As regards the dominant M2 constituent, findings indicate AR = 0.992 and PHLD = -0.11°.
Meanwhile, AR for S2 = 0.972 after allowance for the air tide and PHLD = 4.19°. However, findings for the small
diurnal tides were unsatisfactory, yielding historical amplitudes only about 60% of their modern ones, which is less
plausible.
These results are not perfect but provide confidence in the findings using the harmonic methods, at least for the M2
constituent. The sparse Maskelyne data, and its daylight bias, are likely to be the major reasons for what appears to
be the poor performance of the response method for the diurnals.

**4 Comparisons of Cartwright's and Our Own Analyses**

Cartwright (1971) used a complicated form of response method, allowing for measurements at arbitrary fractions of
an hour so as to accommodate the irregular times of Maskelyne's measurements, and using data from Ascension
Island as a reference record in the method. Unfortunately, those Ascension data are no longer available and there are
details of his method which are hard to follow. Therefore, it is impossible to reproduce Cartwright (1971) in all
respects. The best we can do at this stage is to make our own analyses and see if our main conclusions agree with
his.
Cartwright (1971) decided that the 1761 M2 tide amplitude was 0.98 of the modern value (or 0.984 by a different
method) and that the historical phase lag was 2.9 ° (or 2.39 ° by the different method) less than now. We present
these in Table 4 in the same form as Table 2.[5] These are almost the same as the conclusions for M2 in Section 3.3
when allowing for a datum shift or not. They are also consistent with the M2 findings in Section 3.4. Cartwright
concluded that his PHLD values were indistinguishable from zero given the noise in the records.

---

[5] We believe we have the correct signs for PHLD in this table: Cartwright chose to work with phase leads rather than lags.



**Table 4:** Values of ratio of historical to modern amplitudes (AR) and historical phase lag minus modern phase lag (PHLD, deg) obtained in the two methods of Cartwright (1971).

|  | M2 | | S2 | | K1 | | O1 | |
|---|---|---|---|---|---|---|---|---|
|  | AR | PHLD | AR | PHLD | AR | PHLD | AR | PHLD |
| Method 1: | 0.98 | -2.9 | 1.02 | 2.9 | 1.0 | 13.0 | 0.83 | 13.0 |
| Method 2: | 0.984 | -2.39 | 1.016 | 2.39 | 0.95 | 9.1 | 0.95 | 9.1 |

In the case of S2, he stated that 'the trends are almost exactly reversed [compared to M2]'. Cartwright used a pressure sensor for his 1969 measurements, as we did later for STHL2 and STHL4, but his 1971 paper makes no mention of the complication of the S2 air tide. However, the amplitude given for S2 in his Table 2 is 11.2 cm which indicates that, for one reason or other, he believed that the amplitude of S2 in the ocean tide at this location was essentially the same as we have used in STHL4.X, as discussed in Section 3.2.

In the present study, the first harmonic method of Section 3.3 showed that, if one allows for the datum shift or not, then the 'reverse S2 trend' is indeed the case for phase lag, although the historical S2 amplitude is a little smaller than today, as for M2, rather than a little larger as Cartwright obtained. That is the same conclusion as for the second harmonic method in Section 3.4 once the air tide is allowed for. It cannot be important to agonise about the very small changes in S2 amplitude. That would require unreasonable assumptions concerning the accuracy of Maskelyne's measurements: for example, on the accuracy of the 3-inch graduations of the tide pole and an assumption that it was perfectly vertical. In addition, there was the inevitable complication of making accurate tidal measurements in the frequent presence of high surf. Similar to Cartwright, we do not believe there is any significance in the 'reverse trend' for S2 phase lag, given the variability in the records, consistent with PHLD near zero for both of the main semidiurnals.

The diurnals are more problematical. Cartwright (1971) considered that the historical amplitudes of the diurnals were 1.0 and 0.85 times the present ones for K1 and O1 respectively with both historical phase lags about 13 ° larger. In a second method, he considered both historical amplitudes to be about 95% of the present ones with historical phase lags 9 ° larger.

The phase lag findings of Cartwright (1971) are consistent with those of the first harmonic method in Section 3.3 when allowing for a datum shift, with phase lags larger in historical times (although not by as much as 13°). On the other hand, the first method has smaller/larger historical amplitudes for K1/O1, compared to similar or slightly smaller historical amplitudes for both constituents in Cartwright (1971). If one does not allow for a datum shift, the first method again suggests smaller/larger historical amplitudes for K1/O1 but PHLD values moving in opposite directions unlike in Cartwright (1971).

The conclusions on the diurnals from the second harmonic method of Section 3.4 are consistent with Cartwright (1971), in there being historical amplitudes a few percent smaller than today, although the phase lags for K1 and O1 move in opposite directions in the second method unlike in Cartwright (1971). One notes the large Mm obtained in Section 3.4 from the interpolated Maskelyne data is consistent with the datum shift considered in Section 3.3. Our own attempt at response analysis in Section 3.5 supported the case for essentially unchanged semidiurnal tides.

Some differences in findings between this report and Cartwright (1971) are to be expected for several reasons. One important aspect concerns the data sets that he had available, which were a year of high and low water levels at Ascension in 1958-9, 39 days of his own measurements at St. Helena in 1969, and of course Maskelyne's measurements in 1761. On the other hand, we have about three decades of continuous sea level measurements from St. Helena, acquired through South Atlantic programme of the National Oceanography Centre (Spencer et al., 1993). Those records, from which STHL4.X and STHL2 were derived, are much higher quality than Cartwright's.

Second, the methods used to analyse the data are different. Cartwright used the Ascension record as a reference in a response analysis involving his 1969 data and Maskelyne's data. On the other hand, we have used two forms of harmonic analysis and our own response method. Less important, it seems that Cartwright did not include in his analysis the several measurements that Maskelyne flagged as suspect, whereas we have used all 478 of Maskelyne's measurements in our analyses.

Third, there is the question of whether or not there really was a datum shift on 4 December 1761. Cartwright (1971) did not believe there was any evidence for a 'sensible change in datum', but the present work has shown that there





probably was a small shift. It is possible that the filters Cartwright used in his response method resulted in his analysis
being less sensitive to a small shift. However, that seems not to be the case when using the harmonic method.
Although findings on changes in the main semidiurnal tides (M2 and S2) are largely unaffected, consideration of the
shift does have an impact on findings for the diurnal tides. Unfortunately, the shift happens in almost the middle of
the 40 days of measurements and splitting the data sets into two and analysing them separately, as one might do with
a longer record, is not a suitable option.

**5 Conclusions**

To sum up, the headline results of Cartwright (1971) were that the semidiurnal tides had not changed at St. Helena
since 1761, that the amplitudes of the diurnals were on balance slightly smaller than today, and that they had about
10° larger phase lag in 1761. Both of our new analyses agree qualitatively with those conclusions for the
semidiurnals: we believe historical and modern M2 to be essentially the same and that the S2 amplitude was a couple
of percent smaller in historical times. On the other hand, one notes big differences between methods in Tables 2 and
4 for the diurnals which result from the difficulties of analysing the short Maskelyne data set. As a result, we would
be more hesitant to claim any changes at all in either the semidiurnal or diurnal tides. In retrospect, one wonders
why Cartwright chose to focus on the apparent changes in phase of the diurnals that he obtained, given that his own
discussion on top of p619 shows that their approximately 10° phase lag difference was only a 2-sigma effect.
The findings of Cartwright (1971) were important ones that have been referenced in major reviews of "tides a-
changin" (e.g. Haigh et al., 2020). However, from the perspective of 'old records for new knowledge', his study also
serves as an important example that electronic data sets can be as much at risk as paper records. His own
measurements at St. Helena in 1969 can no longer be found, while his version of the 1958-9 high and low water
record at Ascension and, of course, the Maskelyne data at St. Helena are also missing.[6] Therefore, it is good at least
that a data set of the historical St. Helena measurements made by Maskelyne in 1761 is once again available for any
researcher to investigate. It has been interesting to make our own analyses from that recovered data set, confirming
Cartwright's main findings on the similarity of the predominant M2 constituent in the historical and modern data
from St. Helena.

**6 Data availability**

The small file 'maskelyne_data' referred to above can be accessed via a DOI from https://doi.org/10.5285/e0f2b6ea-
d11d-3102-e053-6c86abc073ab. Its citation is shown here as Woodworth and Vassie (2022). The file has also been
deposited with the British Oceanographic Data Centre in which it has Accession Number POL200133. Any
information on the tidal analyses made in this study may be obtained from both authors.
**Author contributions.** PLW undertook the data rescue aspect of this work. Both PLW and JMV performed the data
analyses and prepared the manuscript.
**Acknowledgements.** We thank Richard Ray and David Pugh for comments on an earlier version of this article.
Richenda Houseago-Stokes of the British Oceanographic Data Centre is thanked for setting up the DOI for the
Maskelyne data set.
**Competing interests.** The contact author has declared that neither author has any competing interests.
**ORCID**
P.L. Woodworth http://orcid.org/0000-0002-6681-239X
J.M. Vassie http://orcid.org/0000-0002-9906-1019

---

[6] Cartwright obtained the 1958-9 Ascension high and low water data from the US Coast and Geodetic Survey (USCGS) so it is probable that a version of this data set is archived by a US centre. Cartwright (1971) also mentions a month of hourly sea levels from Ascension in 1959 that was obtained from the USCGS. The latter does not appear to have been used in the Cartwright (1971) study; if required, a manuscript copy of that short record may be obtained from the present authors.





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
