# Peer review of "Reanalyses of Maskelyne's Tidal Data at St. Helena in 1761 3 4 5 Philip L. Woodworth1 and John M. Vassie1 6 7 1 National Oceanography Centre, Joseph Proudman Building, 6 Brownlow Street, Liverpool L3 5DA, United 8 Kingdom 9 10 Correspondence to: P.L. Woodworth (plw"

_Earth System Science Data, 2022_

## Author Response (AR2)

We enclose below the comments from the office, editor and reviewers. Their paragraphs have a leading hyphen. Our responses are shown beneath each comment.

============================================================================

Comments from the office:

- For the next revision, please merge

   the Figures 3(a), 3(b), 3(c) into one Figure 3 with panels (a), (b), (c).

   the Figures 4(a) and 4(b) into one Figure 4 with panels (a) and (b).

This has been done.

============================================================================

EC1: 'Comment on essd-2022-218', Giuseppe M.R. Manzella, 08 Jul 2022

- The paper proposes an analysis of historical data that can help to understand changes in the characteristics of the tides over the centuries (if changes are occurring). The critical analysis of historical data is very well done and the methods used to verify the reliability of the data and their current value are exhaustive.

- There are some missing details on 'methodologies and technologies' that may be added to satisfy the reader's curiosity.

- In the introduction one could have expected a brief presentation of why tidal characteristics can change (lines 30 - 33) and whether the reasons for the changes are measurable over a period of a few hundred years.

The wording has been expanded slightly to mention changes in water depth as one reason for changes in the tide. However, this is a big subject and the Haigh et al. (2020) reference given here contains a full discussion of possible processes.

- Line 78: … agreed consistently to better than half an inch (12 mm). Considering that the tidal movements are of lesser amplitude than those due to wave movements and other high frequency

coastal movements, I wonder that short waves, spray, shower … may have influenced the measurement reading. May be the pole was in a sheltered place. Are there any indications in the original manuscript or in other works by Maskelyne?

Waves will certainly be much larger than half an inch of course, but being discussed here is the ability of an observer to estimate the 'still water level' at any time visually i.e. to average over the waves during say a minute. In relatively calm conditions one can certainly do that to that accuracy with practice, so I am not surprised that Maskelyne and Mason, who will have had a lot of practice by then, agreed on their measurements. We have done enough tide pole measurements ourselves to appreciate that. One cannot do anywhere as well in a storm of course. We have added a footnote to confirm that and a reference.

- Line 112: … times set down are exact to the minute It would be useful to know the type of clock used exactly to the minute and what is the meaning of 'apparent time' in figure 1.

We have modified the wording at the start of section 3.1 to make what is meant by apparent time clearer. Local apparent time is the time one measures with a sun dial as opposed to mean time which one measures with a clock - the difference, which occurs because the sun is not always on the equator, can be as large as 15 minutes at certain times of the year. This is called the 'equation of time'.

- Line 492: … several measurements that Maskelyne flagged as suspect Is it possible to add more details on how Maskelyne flagged 'suspect' data?

We have altered the wording here to be consistent with that in the header of the data file. Maskelyne flagged some measurements as 'doubtful' and 'very doubtful' because of swell conditions.

- It is a pity that some documents are available only in English libraries. Their on line availability would have fascinated many readers to the subjects of historical oceanography.

We agree on this general remark (and that in RC1). We are ourselves often faced with travelling to London or Cambridge to find a document. However, in the present case we are discussing data that was included at the end of a published Phil Trans paper. The Phil Trans originals will of course be at the Royal Society but it is good that we have access to the historical pdfs.

- However, the article represents an excellent example of revisiting and processing historical data

Many thanks for these remarks.

===============================================================================
======

RC1: 'Comment on essd-2022-218', Giuseppe M.R. Manzella, 09 Jul 2022

The comments have been sent as Chief Editor (EC1) (answer to them)

- I am directly interested to historical oceanography and to the availability of original manuscripts. Most of cited references are in libraries located in UK. It would be interesting to have an open access to the old publications.

Please see our reply under EC1.

================================================================================
======

RC2: Comments by Laurent Testut

We are grateful to Dr. Testut for these comments. It is clear that he must have taken the time to look at the data set in detail, as well as reading the draft.

- This new dataset is undoubtedly of great interest especially in the recent context where ocean tide evolution at the coast but also globally is in discussion in the ocean tidal community. Datasets back to the eighteen century is very rare especially in the Southern Hemisphere making the release of this new old-dataset even more valuable. The sea level data are analysed using 3 different methods which gives confidence on the robustness of the main conclusion of no significant semi-diurnal tide evolution during the past 2 and a half century. This paper also offers nice recipes to help the tidal scientist to process their old short-term datasets using these different methods. Although I mainly agree with the way the data is processed I have some questions listed below :

- (1) The dataset used to estimate the modern tidal constituents (TC), is the sub-surface pressure (SSP) recorded at Saint-Helena between oct-1995 and oct-1996. There is a more recent dataset at the same location, with direct observations of sea level from a radar made between jan-2011 and feb-2013. This dataset is available at University of Hawaii Sea Level Center (as the one used by the authors from 95-96). What drives the choice of analysing the SSP dataset instead of the more recent one ? The recent radar data gives at the mm level the same values as the (S2,S1) modified TC used by the authors without the need to modified S1 and S2.

The reason for using STHL4 (or STHL4.X) set, which was derived from a year of data Oct 1995 - Oct 1996, and STHL2, which was derived from data Nov 1993 - Feb 1995, was that we two authors had looked separately at these spans of data before and were happy with their tidal information content.

As we mentioned, these constants were similar to each other and also to a set derived from 1993-2006 by Richard Ray for an earlier study of St. Helena data. And, as the reviewer says, we could also have used a set derived from later radar data as well. In practice, it doesn't matter much which set is used, but as he says if you use the sets derived from SSP data you have to clearly explain what you have done, which we have. Incidentally, the University of Hawaii data are simply copies of the records at NOC as the gauges at St. Helena were ours.

- (2) In the second approach, they interpolate the Maskelyne data at 1 minute using a tidal prediction. As they authors quoted themselves, there is a clear danger about this method due to a leak of information from the prediction to the observation. Anyway, to make the tidal prediction they use the TC obtained with another dataset (called STHL2 obtained from SSP between Nov-1993 do Feb-1995). As they quote the STHL2 TC is very close to STHL4.X. I don't understand the reason behind using another set of TC for this second approach ? Using the prediction from the STHL4.X would have give almost the same results no ?

Yes, this is right. The reason is that, as mentioned above, we two authors had our favourite sets of data which we were happy with as to their tidal information. We could indeed have used just the one set as far as this paper goes, although by using two sets gives some feel for the stability of present-day constants, which can be represented as a benefit. We have put in a footnote to the paper to explain this.

- (3) The standard deviation of the 478 historical data (28.41 cm) is higher than in any of the recent 2 months period available (from 1995 to 2013 where maximum value is 27.1 cm and the mean standard deviation of every 2 months period is close to 25.5 cm). Does the authors have any explanation of this slightly larger standard deviation. Can it comes from the sampling of the Maskelyne data ? Can it be a slight scale error on the tide staff build by Maskelyne or a small difference in the conversion between inch (used by Maskelyne) and meter (change in definition between 1761 and the official conversion).

We do not find it surprising that an historical data set is perhaps a bit noisier than a modern one, although we do not have a fuller explanation. It could indeed have a contribution from sampling, one could look into that by sampling modern data similarly, but honestly we do not see what point that would be. It is reasonable to warn against possible scale errors in tide pole readings and we have added a few words to a footnote about this.

Minor comments :

- I recommend to add in the text or in a separate table the list of constituents used in the tidal prediction (for the STHL2 and STHL4.X), this will allow the futur users of the Maskelyne dataset to reproduce the results of this paper or at least to test their own harmonic analysis code.

We agree. We have provided a Supplementary Material file listing the full set of STHL4, STHL4.X and STHL2 constants.

- A typo has been found in the Maskelyne data file provided by the authors on line 121 (date 1761-11-16 6:44 PM) a decimal '.' is missing, one should read 11.3 instead of 11 3. This is confirmed in the original Maskelyne report.

Thanks for pointing this out. It makes no difference to the findings reported in the paper (at least to within several decimals). But BODC have set up a new doi for a replacement version of this file, this time with the missing dot, and this new doi refers back to the old one as being a replacment and vice versa.